# Electrodeposition of Co*_x_*NiV*_y_*O*_z_* Ternary Nanopetals on Bare and rGO-Coated Nickel Foam for High-Performance Supercapacitor Application

**DOI:** 10.3390/nano12111894

**Published:** 2022-05-31

**Authors:** Seyedeh Mozhgan Seyed-Talebi, Mohsen Cheraghizade, Javad Beheshtian, Chun-Hsiao Kuan, Eric Wei-Guang Diau

**Affiliations:** 1Department of Applied Chemistry, National Yang-Ming Chiao Tung University, Hsinchu 300093, Taiwan; 2Advanced Surface Engineering and Nano Materials Research Center, Department of Electrical Engineering, Ahvaz Branch, Islamic Azad University, Ahvaz, Iran; 3Department of Chemistry, Shahid Rajaee Teacher Training University, Tehran, Iran; j.beheshtian@gmail.com; 4Center of Emergent Functional Matter Science, National Yang-Ming Chiao Tung University, Hsinchu 300093, Taiwan

**Keywords:** electrodeposition, nickel foam, Co*_x_*NiV*_y_*O*_z_* ternary nanocomposite, supercapacitor

## Abstract

We report a simple strategy to grow a novel cobalt nickel vanadium oxide (Co*_x_*NiV*_y_*O*_z_*) nanocomposite on bare and reduced-graphene-oxide (rGO)-coated nickel foam (Ni foam) substrates. In this way, the synthesized graphene oxide is coated on Ni foam, and reduced electrochemically with a negative voltage to prepare a more conductive rGO-coated Ni foam substrate. The fabricated electrodes were characterized with a field-emission scanning electron microscope (FESEM), energy-dispersive X-ray spectra (EDX), X-ray photoelectron spectra (XPS), and Fourier-transform infrared (FTIR) spectra. The electrochemical performance of these Co*_x_*NiV*_y_*O*_z_*-based electrode materials deposited on rGO-coated Ni foam substrate exhibited superior specific capacitance 701.08 F/g, which is more than twice that of a sample coated on bare Ni foam (300.31 F/g) under the same experimental conditions at current density 2 A/g. Our work highlights the effect of covering the Ni foam surface with a rGO film to expedite the specific capacity of the supercapacitors. Despite the slightly decreased stability of a Co*_x_*NiV*_y_*O*_z_*-based electrode coated on a Ni foam@rGO substrate, the facile synthesis, large specific capacitance, and preservation of 92% of the initial capacitance, even after running 5500 cyclic voltammetric (CV) scans, indicate that the Co*_x_*NiV*_y_*O*_z_*-based electrode is a promising candidate for high-performance energy-storage devices.

## 1. Introduction

Production of electricity without greenhouse gas emission, mostly in the form of CO_2_, from abundant renewable energy sources such as solar, wind, tidal sea energy, and geothermal power is a challenging task because the supply does not correspond to demand [1,2]. Furthermore, effective storage of the electricity generated from energy sources in a stand-alone system requires a powerful battery or supercapacitor or another energy-storage device [3,4]. There is thus a critical time for researchers to satisfy the industrial demands of more efficient energy-storage devices using favorable, clean, cheap, and environmentally friendly electrode materials [5,6,7]. In the field of electrochemistry, an electrochemical capacitor (EC, also known as “supercapacitor”) is a device that stores the generated energy from reactions between the electrode materials and an electrolyte. Therefore, the term of “supercapacitor” is used to describe the properties of an electrode material that behaves like a capacitor in its electrochemical signature. Supercapacitors have become favorable candidates in recent years for systems that store energy from reactions between electrode materials and an electrolyte [8,9]. Much effort has been exerted to improve the energy-storage performance of both electrochemical supercapacitors and pseudocapacitors using transition-metal oxides as promising electrode materials [10,11,12,13], but their poor electrical conductivity, small power density, poor specific capacitance, unsatisfactory cycle life, great cost, weak environmental safety, and large volume modifications greatly hinder their commercial applications [14]. From the perspective of interface engineering, the electrochemical performance of a supercapacitor is directly related to defects on the surface of the electrode materials that influence the electrical conductivity, chemical reactivity, selectivity of ions, and specific surface area of the electrode [15]. Two- and three-dimensional boundaries at an interface between electrolyte and electrodes can be considered active sites for chemical reactions on the surface of the electrodes, so that rational design and engineering of the surface of the electrodes are critical to improve the performance of devices [16,17]. The highly conductive Nickel foam has been utilized in previous research as a low-cost scaffold for the uniform growth of electrode materials and is also used for facilitating electron transport and rapid migration of electrolyte ions by making the ion diffusion paths shorten [18]. A simple strategy to improve the conductivity of the electrode materials is to deposit them on graphene-based materials as highly conductive substrates [19,20,21]. With large surface areas, great conductivity, and excellent mechanical properties, reduced graphene oxide (rGO) makes a conducting path more favorable and facilitates a rapid electrochemical kinetic process during charge or discharge at a large current density. The scientific design of electrode materials on a rGO-coated substrate can improve electrochemical performance by decreasing the ion aggregation on the surface, facilitating ion migration, balancing the interface [22,23], and offering an efficient electron transport path. By taking the advantages aforementioned, we employed rGO-coated nickel foam (Ni foam) as a suitable substrate to design a high-performance cobalt nickel vanadium oxide (Co*_x_*NiV*_y_*O*_z_*)-based electrode using the electrodeposition method. The electrodeposition method was selected among the different available deposition methods due to its various advantages for energy storage devices such as low temperature, large scale, vacuum-free, and good control of the deposition parameters [24,25,26,27]. In a Co*_x_*NiV*_y_*O*_z_*-based ternary nanocomposite, Co element provides increased electronic conductivity, and Ni element offers a large capacity and can improve the active site density. V element possesses excellent electrical conductivity due to its large theoretical capacitance, wide potential windows, and satisfactory electrochemical performance that can result in improved capacitive performance of a device [28,29,30,31]. The incorporation of varied metal ions might yield multi-phase materials and introduce abundant structural defects, which can be favorable for electrochemical energy storage.

Most of the published works overlook or even ignore the effect of graphene covering of Ni foam substrate on the ratio of surface control and diffusion control in mechanism of charge–discharge in supercapacitors. By using a comparison method, we systematically studied such effects on the charge–discharge mechanism in Co*_x_*NiV*_y_*O*_z_*-based electrode material. Our results showed that surface control has more effective impact than difusioun control in supercapacitor behavior of Co*_x_*NiV*_y_*O*_z_*-based electrode.

## 2. Materials and Methods

### 2.1. Chemicals

Cobalt(II) sulfate heptahydrate (CoSO_4_·7H_2_O; ≥99% purity, CAS No: 10026-24-1, EC number: 233-334-2), nickel sulfate hexahydrate (NiSO_4_·6H_2_O; CAS No: 1010-97-0, EC number: 232-104-9), vanadium pentoxide (V_2_O_5_; 99.95%, CAS No: 1314-62-1, EC number: 215-239-8), sodium hydroxide (NaOH; CAS No: 1310-73-2, EC number: 215-185-5), sodium nitrate (NaNO_3_; 99.99% Suprapur, CAS No: 7631-99-4, EC number: 231-554-3), propanone (CH_3_COCH_3_; ≥99.5%, CAS No: 67-64-1, EC number: 200-662-2), potassium hydroxide (KOH; CAS No: 1310-58-3, EC number: 215-181-3), and hydrochloric acid (HCl; 37%) were purchased from Merck (Darmstadt, Germany). The Ni Foams (100 × 100 mm^2^; thickness 1.6 mm; pore size 200–300 μm) were purchased from MTI (Seoul, South Korea). All deionized water used in the synthesis procedure was obtained from our in-house Merck Millipore water purification system (Darmstadt, Germany).

### 2.2. Material Characterization

The surface morphology and elemental composition of the prepared electrodes were characterized with a field-emission scanning electron microscope (FESEM, MIRA3 TESCAN, Brno, Czech Republic) equipped with an energy-dispersive X-ray (EDX) analyzer. X-ray photoelectron spectra (XPS) were recorded with a Surface Analysis machine (Thermo Fisher K-ALPHA, Waltham, MA, USA). Functional groups and chemical bonding of deposited ternary composites were identified with Fourier-transform infrared (FTIR) spectra (Perkin Elmer, Waltham, MA, USA) in wavenumbers 4000–400 cm^−1^ employing KBr pellets. The specific surface area (SSA) of substrates was investigated with Brunauer, Emmett, and Teller (BET), Barrett, Joyner and Halenda (BJH) methods using N_2_ adsorption–desorption tests (Micrometrics ASAP 2020, Norcross, GA, USA).

### 2.3. Electrochemical Instruments and Measurement 

To measure the potential of prepared electrodes in highly efficient supercapacitors, all quantitative testing methods, including cyclic voltammetry (CV) and galvanostatic charge–discharge (GCD) tests of electrodes, were performed using an electrochemical analyzer (Autolab PGSTAT-204, Metrohm, Kanaalweg, The Netherlands). All electrodeposition processes and electrochemical measurements of separate electrodes were recorded near 300 K in a homemade deposition bath; the distances between conventional silver/silver chloride (Ag/AgCl) reference electrode, platinum counter electrode, and Ni foam work electrode were kept constant at 2 cm.

### 2.4. Electrochemical Deposition of Co_x_NiV_y_O_z_ Nanocomposite on Bare Ni Foam and Ni foam@rGO

One efficient strategy to improve the energy-storage capacity of a supercapacitor is to incorporate various binary or ternary metal oxides or hydroxides such as Ni–Co [32,33], Co–V [34], Co–Al [35], Al–Mn–Co [36], Ni–Co–Mn [37,38], Zn–Ni–Co [28], and oxides or hydroxides. The growth of novel 2D nanosheets consisting of oxides of Co, Ni, and V was thereby expected to increase the number of electrochemical sites, resulting in a highly efficient supercapacitor. In a Co*_x_*NiV*_y_*O*_z_*-based electrode, Co element improves the conductivity; Ni element can increase the specific capacity on increasing the active site density. V element offers excellent electrical conductivity due to its wide potential windows and satisfactory electrochemical performance that can result in improved capacitive performance of a device [39]. The optimized electrochemically synthetic procedure of Co*_x_*NiV*_y_*O*_z_* crochet nanosheets beautifully grown on a Ni Foam surface is illustrated in Figure 1; detailed information of all preparation and deposition steps is presented below.

#### 2.4.1. Preparation of Ni Foam Substrate

Before deposition of Co*_x_*NiV*_y_*O*_z_* ternary nanosheets on a Ni foam substrate, Ni foam was cut into pieces 1 × 2 cm^2^ and subsequently sonicated in HCl solution (3 M), absolute ethanol, propanone, and water for 15 min to eliminate the surface oxide layer, contamination, and organic materials from the surface of Ni foam, followed by dehydration in an oven at 70 °C under air for 3 h.

#### 2.4.2. Preparation of the rGO-Coated Ni Foam (Ni foam@rGO) Substrate

Graphene-oxide (GO) films were synthesized with a modified Hummers method that is described elsewhere [39,40]. A typical FESEM image and XRD pattern of the synthesized GO are available in Appendix A. To prepare a GO suspension (5 mg/mL), synthesized GO (250 mg) was dispersed in de-ionized (DI) water (50 mL) with magnetic stirring for 2 h followed by sonication for 1 h in an ultrasonic bath. The cleaned Ni foam was soaked in a GO suspension for 1 h and purged with N_2_ for 15 min to remove oxygen from the reaction solution. After applying a simple cyclic voltametric electrochemical treatment, we deposited the rGO films on the Ni foam substrates in a voltage range −1.5 to −0.5 V on applying 20 cycles at sweep rate 200 mV/s; a color change of the Ni foam substrate was visually observed. The XPS spectrum of rGO coated substrate shown in Figure 2 revealed that, under a negative applied voltage, GO converted to rGO [41,42,43]. The prepared Ni foam@rGO substrates were rinsed with DI water several times and dried at 60 °C in an oven overnight.

According to measurements of the nitrogen adsorption/desorption isotherm and plots of the distribution of pore size of pure Ni foam and Ni foam@rGO substrates (Appendix A), the samples exhibited an adsorption/desorption isotherm type IV. Because of the presence of varied rGO sheets, N_2_ molecules can absorb on both sides of a rGO surface. The specific surface area (SSA) of resultant layer (Ni foam@rGO substrate) is thereby expected to become much larger (1100 m^2^/g) than the pure Ni foam substrate (286 m^2^/g) [44]. Plots of the corresponding distribution of Barrett–Joyner–Halenda (BJH) pore size indicated distributed pores of size about 6 nm; plots of the distribution of pore size of the Ni foam@rGO substrate indicated the formation of pores about 8 and 35 nm, because of the interspaced pores of separate rGO layers. The above evidence reveals that, by coating rGO onto the surface of Ni foam, the SSA of the substrate became larger, which greatly improves the specific capacitance of a coated nanocomposite on a surface of the Ni foam@rGO substrate.

#### 2.4.3. Electrodeposition of Nanocomposite on Bare and rGO-Coated Ni Foam Substrates

The Co*_x_*NiV*_y_*O*_z_* ternary nanocomposite was directly grown on the Ni foam substrate with a facile one-pot electrodeposition method in a typical three-electrode deposition bath. The cleaned Ni foam served directly as a working electrode; a Pt plate and Ag/AgCl (in saturated KCl) were employed as counter and reference electrodes, respectively (Figure 1). The aqueous solution consisting of cobalt sulfate heptahydrate (Co(SO_4_)·7H_2_O), Nickel sulfate hexahydrate (NiSO_4_·6H_2_O), vanadium pentoxide (V_2_O_5_), and NaNO_3_ in molar ratio 1.5:1.5:2:3 was prepared as a precursor solution for the electrodeposition of petal-like Co*_x_*NiV*_y_*O*_z_* nanosheets on the surface of a Ni foam substrate. The mixture was dispersed ultrasonically in ultrapure deionized water (50 mL) after heating to 55 °C for about 30 min to give a yellowish-orange solution with no suspended solids. The homogeneous precursor was placed in a beaker; the electrodeposition was conducted in a home-made deposition bath with distances between the Pt and Ag/AgCl electrodes kept at 2 cm. The cleaned Ni foam was immersed in a suspension and purged with N_2_ for 15 min to remove oxygen from the reaction solution. This process resulted in a uniform aqueous dispersion of the precursors and a uniform layer remaining on the surfaces of the Ni foam. The surface area of Ni foam exposed to the deposition was 1 cm^2^ (1 × 1 cm^2^); the duration of deposition was 600 s.

The electrochemically prepared electrodes were rinsed with DI water several times and dried at 80 °C in an oven overnight, and then calcined in a furnace at 300 °C for 2 h without further treatment. The furnace temperature was increased from 30 °C to 300 °C to initiate a uniform propagation of metal oxides with networks of nickel ligaments. On heating samples to 300 °C, the deposited metal hydroxides (Ni(OH)_2_, VOH, and Co(OH)_2_) on the surface of Ni foam were converted to metal oxides (NiO, V_2_O_5_, and Co_3_O_4_) to generate the Co*_x_*NiV*_y_*O*_z_* crochet nanosheets. The layered oxides are more stable than their corresponding hydroxide layers. After heating for 2 h, the petal-like growth was completed at 300 °C; a thin film of crochet-structured electrode materials was formed that comprised several 2D nanosheets covering the entire area of the Ni foam substrate. The resultant electrode delivered a large capacity because of facile ion migration on its surface, its large surface area for redox reactions, and large specific capacitance of each component in the electrode material. The surface of the electrochemically synthesized non-structural films on the Ni foam substrate was significantly activated with KOH solution (1.0 M) to accelerate the electron transfer and had a large surface area as an ideal electrode material, favoring the performance of rapid redox reactions.

## 3. Results and Discussion

Figure 2a shows a XPS survey scan of rGO-coated nickel foam (Ni foam@rGO) substrate that clearly indicates the presence of the C-1*s*, Ni-2*p*, and O-1*s* signals. The inset table also presents the elemental distribution (at %) of the desired elements. The existence of a Ni signal due to the Ni foam substrate is implied by the formation of a thin layer of rGO. In Figure 2b,c the XPS signals of each individual element are shown and identified. For C-1*s* (Figure 2b), the signals related to C binding with O and H are weak, which is due to some hydroxide groups (–OH) removed from graphene and formation of a rGO layer on the Ni foam substrate. For Ni-2*p* (Figure 2c), the XPS signals of the various Ni bonds, including Ni–O and Ni–OH, were observed. The O-1*s* signal (Figure 2d) also confirmed the binding of O with C and Ni [45,46,47].

The surface morphology of the layer fabricated on the Ni foam substrate was investigated with a FESEM. Figure 3 shows typical FESEM images of the electrodes as prepared with various magnifications. These features reveal a uniform distribution of petal-like Co*_x_*NiV*_y_*O*_z_* nanosheets across the Ni foam surface with sizes in range 200–500 nm and wall thicknesses about 10–25 nm. 

Figure 4 shows the EDX and corresponding elemental mapping images of a typical deposited Co*_x_*NiV*_y_*O*_z_* layer on the Ni foam substrate. Analysis of the surface morphologies and chemical composition mapping of modified Ni foam confirmed a uniform distribution of Co, V, Ni, and O elements on the entire detected area of the electrode surface, indicating a homogeneous surface covered with well-formed petal-like Co*_x_*NiV*_y_*O*_z_* nanostructures.

On presenting rGO onto the surface of the Ni foam, nanocomposites can grow on both sides of the rGO sheets. The FESEM images of a typical interlaced petal-like Co*_x_*NiV*_y_*O*_z_* nanosheet deposited on the Ni foam@rGO substrate (Figure 5) show a large aggregation of petal-like Co*_x_*NiV*_y_*O*_z_* nanostructres coated on rGO for comparison with nanosheets coated on a bare Ni foam substrate in the same condition. The tight structure of nanopetals is expected to decrease the contact resistance of a deposited layer and electrolyte. The larger surface area of the electrode covered by nanopetals leads to increased contact of the inner structure of Co*_x_*NiV*_y_*O*_z_* with an electrolyte during electrochemical measurement, so that it can provide a feasible ion-transport path and decrease the inner-pore resistance.

Figure 5a reveals that rGO sheets did not stick tightly together. During the growth of Co*_x_*NiV*_y_*O*_z_* on the surface of rGO coated Ni foam substrate, the nanopetals grow on both side of rGO sheets, acting as spacers to prevent the restacking of rGO sheets and form a porous three-dimensional (3D) structure. The resultant morphology is most effective in facilitating the penetration of electrolyte within electrode material and enhancing the super capacitor efficiency. This approach improves the internal binding and decreases the resistances between active materials and Ni foam current collectors’ substrate. 

Figure 6 and Figure 7 present XPS of the samples. The XPS origins are specified according to the literature [45,48,49,50,51]. The survey scan in Figure 6a for a Co*_x_*NiV*_y_*O*_z_* sample coated on Ni foam indicates the existence of the desired elements in corresponding binding energies (the inset table also presents the elemental distributions). Ni shows only oxidation state Ni^2+^ (Figure 6c), but Co and V present varied oxidation states (Figure 6b,d). These results indicate the formation of an oxide phase for elements Co, Ni, and V, confirming the formation of the Co*_x_*NiV*_y_*O*_z_* nanostructure. The O-1*s* binding energies (Figure 6e) also confirm the existence of Co, Ni, and V in oxide states.

The survey scan of a Co*_x_*NiV*_y_*O*_z_* nanocomposite coated on a Ni foam@rGO substrate also presented the desired elements with the corresponding elemental distribution. Like the previous samples, the formation of rGO and Co*_x_*NiV*_y_*O*_z_* materials is evident (Figure 7b–f).

The accurate identification of the different structural defects in the Co*_x_*NiV*_y_*O*_z_* system is very difficult and complicated. However, overall, the metal-oxide compounds (which commonly show polycrystalline semiconducting behavior) usually show O vacancy (O_v_) as a common and dominant structural defect [52]. According to this and considering the XPS results, the O content of the Co*_x_*NiV*_y_*O*_z_* system at the Ni Foam@rGO substrate was increased. Therefore, the O_v_ can decrease. The lower defective Co*_x_*NiV*_y_*O*_z_* system can act as a more capacitive electrode material. The obtained results also verified this trend. When the Co*_x_*NiV*_y_*O*_z_* system was biased by an external charge source, the injected charges initially passivated the defect state [53,54] and then charge accumulation occurred on the electrode surface. Therefore, the lower defective electrode can show the more capacitive behavior. This relation and outcome are in good agreement with the obtained results in the present research.

The electrochemically prepared working electrodes were analyzed with cyclic voltammetry (CV). The comparative CV curves of the deposited electrode materials on the bare Ni foam and Ni foam@rGO substrates in KOH aqueous electrolyte (1.0 M) were initially characterized in a potential window −0.2 to 0.5 V (vs. Ag/AgCl) at scan rates varying from 5 mV/s to 200 mV/s. Comparing the CV curves of bare and rGO-coated Ni foam substrate (Appendix A), and the CV curves of prepared working electrodes recorded in Figure 8a,b, the same behavior was clearly observed for prepared electrodes at varied scan rates because of the redox behavior of nickel, cobalt, and vanadium ions. The overall redox current also increased on increasing the scan rate, representing a rapid charge transport at larger scan rates. 

The charge storage and specific capacitance of electrodes when cycled in an aqueous KOH electrolyte essentially depend on the reactions on the surface of electrode materials, including electrochemical adsorption/desorption of cations and anions at the interface of electrode and electrolyte. Furthermore, the relationship between the redox current and sweep rate is also important. The metal hydroxide mechanism in aqueous KOH electrolyte can be expressed by the following redox equations of the electrode during the CV tests [55,56,57]:(1)$$\mathrm{Ni}{\left(\mathrm{OH}\right)}_{2}+{\mathrm{OH}}^{-}\;↔\mathrm{NiOOH}+H_{2}O+e^{-}$$
(2)$$\mathrm{Co}{\left(\mathrm{OH}\right)}_{2}+{\mathrm{OH}}^{-}\;↔\mathrm{CoOOH}+H_{2}O+e^{-}$$
(3)$$\mathrm{CoOOH}+{\mathrm{OH}}^{-}↔{\mathrm{CoO}}_{2}+H_{2}O+e^{-}$$

The doped V element may also undergo redox reaction as follows [40]:(4)$$V_{2}O_{3}+2{\mathrm{OH}}^{-}↔2{\mathrm{VO}}_{2}\cdot {\mathrm{xH}}_{2}O+H_{2}O+2e^{-}$$
(5)$$2{\mathrm{VO}}_{2}\cdot {\mathrm{xH}}_{2}O+2{\mathrm{OH}}^{-}↔V_{2}O_{5}\cdot {\mathrm{xH}}_{2}O+2H_{2}O+2e^{-}$$

Moreover, the logarithmic dependence of the current on the scan rate was presented in Figure 8c,d for the study of the mechanism of charge–discharge. As we can see, the slope of the fitted line for Co*_x_*NiV*_y_*O*_z_* film on the Ni foam substrate is 0.68. It means that the charge current mainly comes from a solid-state diffusion-controlled intercalation process [58]. The addition of the rGO on the Ni Foam substrate also did not change the charge storage mechanism. The discharge mechanism is also a hybrid process. When the Ni foam@rGO has been used, the slope of the fitted line is reduced in the discharge step. This reduction trend can indicate that the discharge step is more solid-state diffusion-controlled intercalation in comparison to the charge step for Ni foam@rGO substrate. These results show that the charge–discharge contains the diffusion-controlled and surface-controlled processes. Therefore, we can present them by the equation *i*(V) = k_1_*v* + k_2_*v*^1/2^. In this equation, *i*(V) is the anodic/cathodic current, k_1_ and k_2_ are coefficients, and *v* is the scan rate. Based on this, Figure 8e,f were presented. The fitted lines in these figures can be used to obtain the values of k_1_ (slope, which can identify surface capacitive behavior) and k_2_ (intercept, which can identify diffusion-controlled capacitive behavior) [59]. The obtained values for k_1_ and k_2_ indicated that the dominant capacitive behavior of our samples is a surface-controlled process.

Figure 9 presented the typical CV curves and contribution of the different mechanisms on the capacitive behavior of the Co*_x_*NiV*_y_*O*_z_* electrode deposited on the different substrates. It is clear that the diffusion-controlled process has a lower contribution on the capacitive behavior of the Co*_x_*NiV*_y_*O*_z_* electrode. The fact that the charge/discharge curves are straighter further indicates that the nanopetal Co*_x_*NiV*_y_*O*_z_*-based electrode possesses a fast and reversible charging/discharging capability. Furthermore, a larger loop curve area of Co*_x_*NiV*_y_*O*_z_*-based electrode coated on Ni foam@rGO substrate exhibits higher electro-chemical and capacitive behaviors.

The current density of the coated electrode on Ni foam@rGO substrate is greater than for the prepared sample on the Ni foam substrate at the same scan rates. Because of the increased conductivity and supercapacitive behavior of interlaced petal-like Co*_x_*NiV*_y_*O*_z_* nanosheets in the presence of a conducting rGO layer, the obtained CV curves of the coated electrode on Ni foam@rGO show an area at all scan rates greater than for the sample coated on bare Ni foam. According to the BET plots of substrates (Appendix A), the specific surface area of resultant NiO@rGO substrate became much larger than of the pure Ni foam substrate. The greater energy-storage capacity can be attributed to the growth of Co*_x_*NiV*_y_*O*_z_*-crochet nanosheets on a rGO coated surface with increased surface area, and randomly distributed nanosheets across the surface of the coated electrode on the Ni foam@rGO substrate (Figure 5a), which easily connect the inner surface area with the outer electrolyte. This structure hence facilitates rapid ion intercalation and extraction.

The electrochemical tests led to volumetric changes of the layers, as monitored by FESEM and FTIR analysis. The FESEM images (Figure 10) of electrodes before and after electrochemical tests show that the microstructure and morphology of the petal-like Co*_x_*NiV*_y_*O*_z_* nanosheets were modified during the electrochemical tests. The modified structures had absorption of electrolyte ions, causing a decreased specific capacitance after undergoing consecutive CV scans.

Figure 11 displays typical FTIR spectra of various Co*_x_*NiV*_y_*O*_z_*-based electrodes deposited on bare Ni foam and Ni foam@rGO substrates before and after the electrochemical tests. For all samples, the lines at 1634 cm^−1^ and 3400 cm^−1^ are assigned to stretching and deformation vibrations of adsorbed water molecules (hydroxyl (O–H) groups) on the surface of electrodes [60,61]. The spectra also show that the V–O=V bonds (about 500 cm^−1^) exhibit modes from important V=O stretching and edge-shared oxygen bending vibrations of V–O–V species [62]. The stretching vibrations of Co–O and Ni–O result in lines at 559 and 649 cm^−1^, respectively [63]. The lines about 537 and 703 cm^−1^ are also attributed to Ni–OH and Co–OH stretching vibration modes [55], respectively. The strong band about 663 cm^−1^ corresponds to the characteristic vibrational Co–O mode. The main line at ~640 cm^−1^ is assigned to the V–O–V vibration characteristic of six-coordinated vanadium, of which the intensity is greater than of the observed line at ~831 cm^−1^ due to V–O vibration in the spectra. The lines at 990–1008 cm^−1^ are attributed to V=O stretching modes [64].

The FTIR results confirmed the formation of a Co–Ni–V oxide structure deposited on the electrode surface. The lines observed at 1460 and 1628 cm^−1^ in Figure 11b are associated with C–OH vibrations. The stretching modes of carbonyl (C=O) and epoxy groups (C–O) also appear at ~1728 and ~1061 cm^−1^, respectively. As Figure 11a shows, the Co*_x_*NiV*_y_*O*_z_*-based electrodes deposited on a Ni foam substrate before and after electrochemical tests exhibit similar spectra. The intense line about 1537 cm^−1^ of Figure 11b is attributed to the skeletal vibration of the graphene sheets, but those lines ascribed to oxygen-containing functional groups were weakened in FTIR spectra of the Co*_x_*NiV*_y_*O*_z_*-based electrode deposited on the Ni foam@rGO substrate after 5500 consecutive charge–discharge cycles in KOH, revealing that these functional groups were partially removed during reduction; most GO was thus transformed into rGO.

The above FT-IR spectral information supports the formation of Co*_x_*NiV*_y_*O*_z_* nanostructures. The information obtained from FT-IR spectra confirmed the XPS results about the formation of rGO and Co*_x_*NiV*_y_*O*_z_* on the surface of Ni foam substrates. Comparison with the FTIR spectra of electrode materials before electrochemical tests indicated that the decreased intensities of modes related to V, Ni, and Co coated on the surface of the Ni foam@rGO substrate confirm that only certain regions of the electrode material remained on the surface of the Ni foam@rGO, as shown in the FESEM images (Figure 10). The modified intensity of FTIR lines in range 500–2000 cm^−1^ is due to the effective decrease of functional groups on the electrode surface during the electrochemical reduction step in KOH solution.

The specific capacitance (C) of the electrode materials is well estimable from the profile of galvanostatic charge–discharge (GCD) measurements with the following equations,
Cs = ∫(I dv)/(2 s ∆V m)(6)
Cs (F g^−1^) = (I ∆t)/(m ∆V)(7)
in which I denotes discharge current, ∫ I dv implies charge area calculated from integration of the half CV curve (in C), *s* denotes scan rate (mV/s), ΔV represents the voltage window (V), m represents mass of Co*_x_*NiV*_y_*O*_z_* electrode material loaded on the surface of Ni foam (in mg), and Δt is the duration of discharge (s).

We performed GCD tests for the substrates (Appendix A), and coated electrode materials onto various substrates (Figure 12). The nonlinearity of GCD curves at current density varied in the range 2 to 150 A g^−1^ indicates a quasi-reversible ion transfer between electrode materials and electrolyte. The GCD test results revealed that the compound coated on the Ni foam@rGO substrate exhibits improved electrochemical performance, which is consistent with the cyclic voltammetry presented above. A possible reason is the availability of a large portion of the inner surface area with an outer electrolyte that can ultimately serve as an easy ion-transport path to decrease the inner-pore resistance. The results demonstrated that on increasing the current density, the discharge duration decreases. The electrode prepared on the Ni foam@rGO substrate delivered specific capacitances 701.08, 613.42, 584.28, 503.57, 444.85, 348.85, 237.85, 157.50, 25.71, and 17.14 F g^−1^ at current densities 2, 5, 10, 15, 20, 30, 50, 75, 100, and 150 A g^−1^, respectively, within a potential range from -0.2 to 0.5 V during charge–discharge curves based on the total mass of active material (calculated data are listed in Appendix A). The specific capacitance of the electrode coated on the Ni foam substrate is as great as 300.31 F g^−1^ at current density of 2 A g^−1^ but decreases to 2.14 F g^−1^ at 150 A g^−1^. As expected, the electrode prepared on the Ni foam@rGO substrate has a much greater specific capacitance because of the greater surface area than the electrode assembled on a bare Ni foam substrate. The abrupt decrease of capacitance of samples at large current density is related to the decreased electrical conductivity.

The excellent electrochemical stability of Co*_x_*NiV*_y_*O*_z_*-based electrodes arises from the principle of electrochemical energy storage by charge separation at the interface of electrode material and electrolyte, which gives a supercapacitor great advantages over batteries. The capacity retention ratio is an important vital criterion for comparing the cycle life of various electrodes in designs of energy-conversion systems. Figure 12d shows the cycling performance of a typical Co*_x_*NiV*_y_*O*_z_*-based electrode in a three-electrode cell deposited on bare and rGO-coated Ni foam substrates. The samples coated on bare and rGO-coated Ni foam maintained 92.31 and 83.20% of their initial specific capacitance, respectively, at 10 A g^−1^ after 5500 cycles, exhibiting excellent cycling stability of electrode materials coated on bare Ni foam and ensuring its practical application with high performance required in long-term service. The columbic efficiency (*η*) of the charge–discharge cycle was calculated using the below equation [65]:*η*(%) = *t_d_t_c_* × 100(8)

In this equation, the *t_d_* and *t_c_* are the discharging and charging times, respectively. Using the GCD curves (see Figure 12a,b), columbic efficiency was obtained at different currents and presented in Table 1.

Despite higher specific capacitance, the Co*_x_*NiV*_y_*O*_z_*-based electrode material electrodeposited on two dimensional rGO coated Ni Foam failed to achieve the expected stability, mainly due to the high contact resistance between the graphene and Ni Foam substrate. To deal with a similar issue, researchers have made attempts to use other deposition methods such as chemical vapor deposition (CVD) [66], vacuum filtration deposition (VFD) [67], hydrothermal [68], and electrophoretic deposition (EPD) methods [69,70] for generating binder-free electrodes where the graphene material sticks tightly on the metal substrate. Another proposed way to solve this problem is assembling the graphene sheets into a three-dimensional (3D) framework including aerogels, foams, and sponges in graphene-based electrodes [71,72,73,74]. The novelty of this paper is introduction of electrodeposited Co*_x_*NiV*_y_*O*_z_* ternary nanopetals as a new electrode material which provides high performance supercapacitor by enough channels for ion storage and migration. The obtained specific capacitance of the present research was compared with similar reports presented in Table 2. By comparing our results with other metal oxide-based electrode materials fabricated with different deposition methods (some examples are listed in Table 2), our proposed electrode material showed good potential to be used in supercapacitor. Electrochemical deposition as a low-cost, simple controlled, and low temperature growth technique is usually carried out in normal laboratory conditions without requiring a vacuum system [24,25,26,75]. In order to use an easy and rapid growth process, we did not emphasize other deposition methods or preparation of three-dimensional graphene-based substrate to prevent the graphene sheet from restacking and keep their specific surface area to increase the stability of prepared electrode.

## 4. Conclusions

The novel petal-like ternary Co*_x_*NiV*_y_*O*_z_* nanosheets deposited on bare and rGO-coated Ni foam substrates were prepared with a facile, inexpensive, and one-step electrodeposition; their use as a highly efficient supercapacitor is described. With excellent cycle stability (92.31% capacitance retention and 100% columbic efficiency after 5500 cycles) and excellent supercapacitive property, Co*_x_*NiV*_y_*O*_z_*-based electrode materials indicate effective reversibility of charge and discharge on the surface of Ni foam substrates. The proposed novel Co*_x_*NiV*_y_*O*_z_*-based structure with multiple oxidation states for redox reactions, abundant natural resources, and decreased cost and toxicity could also serve as an ideal frame for further deposit as an active material coating, such as commonly used Ni foam. The proposed Co*_x_*NiV*_y_*O*_z_* nanosheets not only provide efficient paths for ion and electron transport, but also allow increased mass loading of active materials. These Co*_x_*NiV*_y_*O*_z_* nanosheets are consequently promising candidates for energy-storage devices. An advantage demonstrated in the presented results is that the rGO-coated Ni foam substrates, which are proposed as an improved conducting option, are slightly less stable than Ni foam-based substrates.

## Figures and Tables

**Figure 1 nanomaterials-12-01894-f001:**
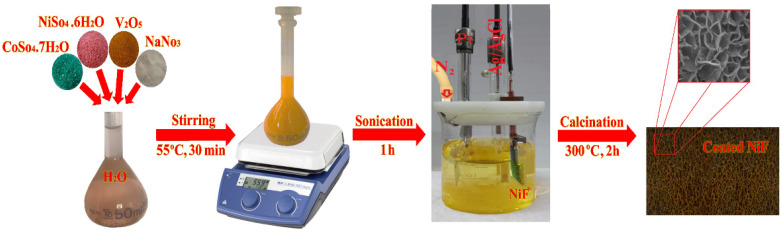
Schematic illustration of electrochemical deposition of Co*_x_*NiV*_y_*O*_z_* ternary nanocomposite on Ni foam substrate: prepared aqueous electrolyte solution, sonicated precursor, and deposition precursors on a Ni foam substrate in a three-electrode deposition cell.

**Figure 2 nanomaterials-12-01894-f002:**
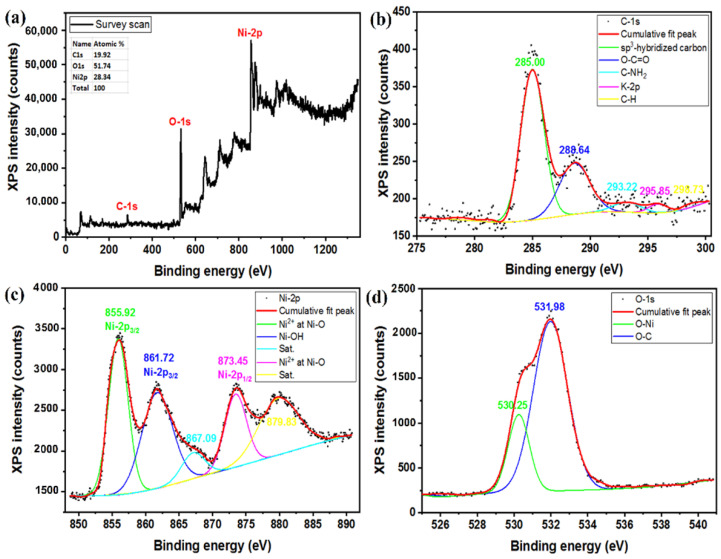
XPS of Ni@rGO substrate, (**a**) survey scan, (**b**) C-1*s*, (**c**) Ni-2*p*, and (**d**) O-1*s*.

**Figure 3 nanomaterials-12-01894-f003:**
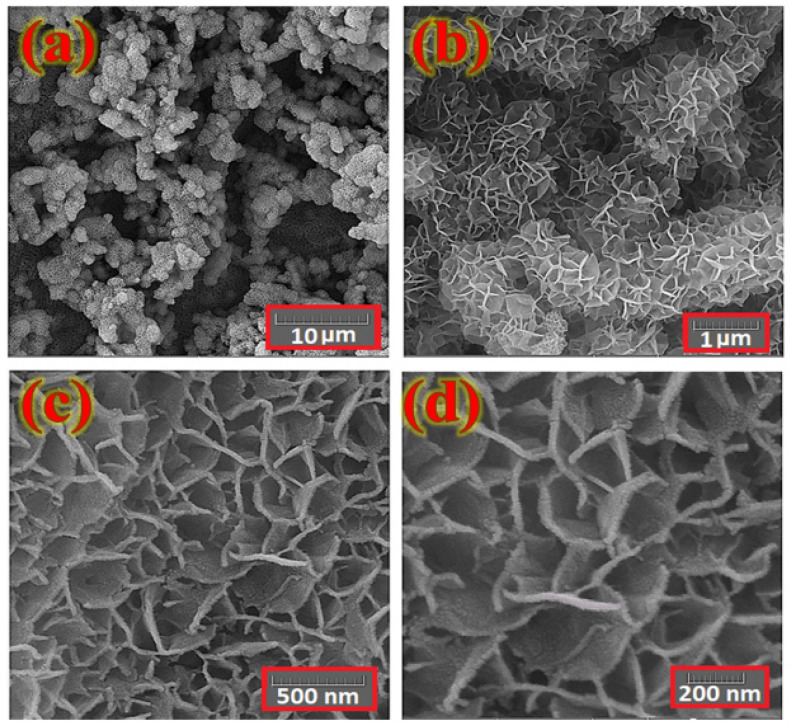
FE-SEM images of a typical Co*_x_*NiV*_y_*O*_z_* layer coated on a Ni foam substrate at various magnifications. (**a**) 10 μm, (**b**) 1 μm, (**c**) 500 nm, (**d**) 200 nm.

**Figure 4 nanomaterials-12-01894-f004:**
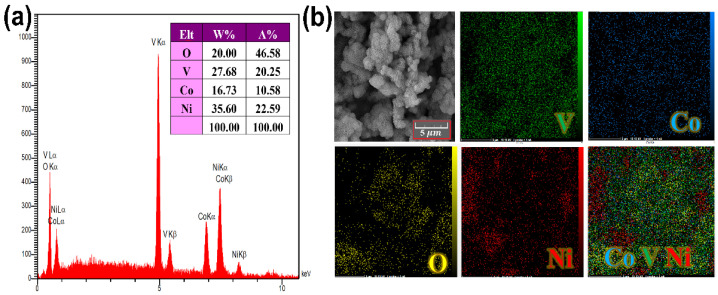
(**a**) EDX spectrum, and (**b**) elemental mapping images of constituent elements of the petal-like Co*_x_*NiV*_y_*O*_z_* coated on Ni foam substrate. V, Co, Ni, and O elements are present on the electrode surface with relative atomic concentrations listed in the inserted table.

**Figure 5 nanomaterials-12-01894-f005:**
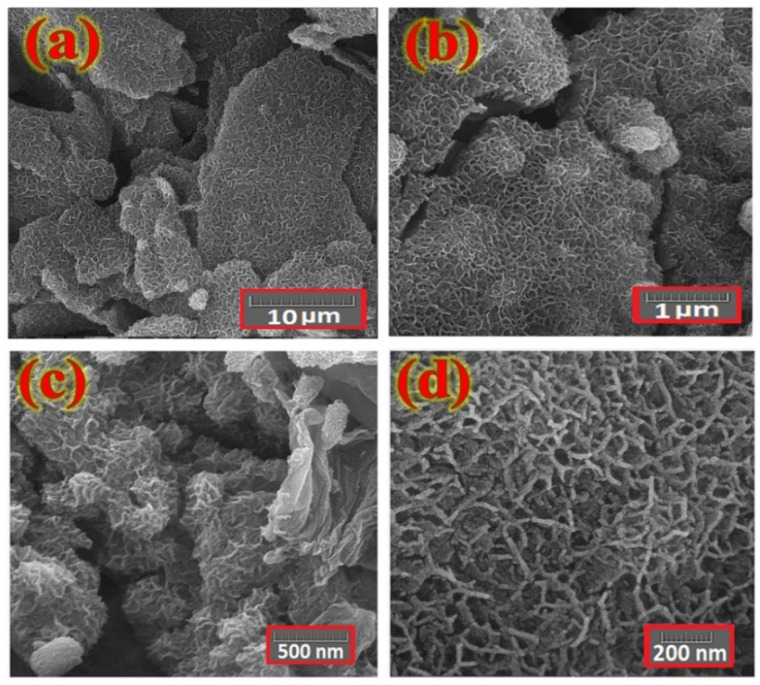
FESEM images of interlaced Co*_x_*NiV*_y_*O*_z_* nanosheets deposited on rGO-coated Ni foam substrate at various magnifications. (**a**) 10 μm, (**b**) 1 μm, (**c**) 500 nm, (**d**) 200 nm.

**Figure 6 nanomaterials-12-01894-f006:**
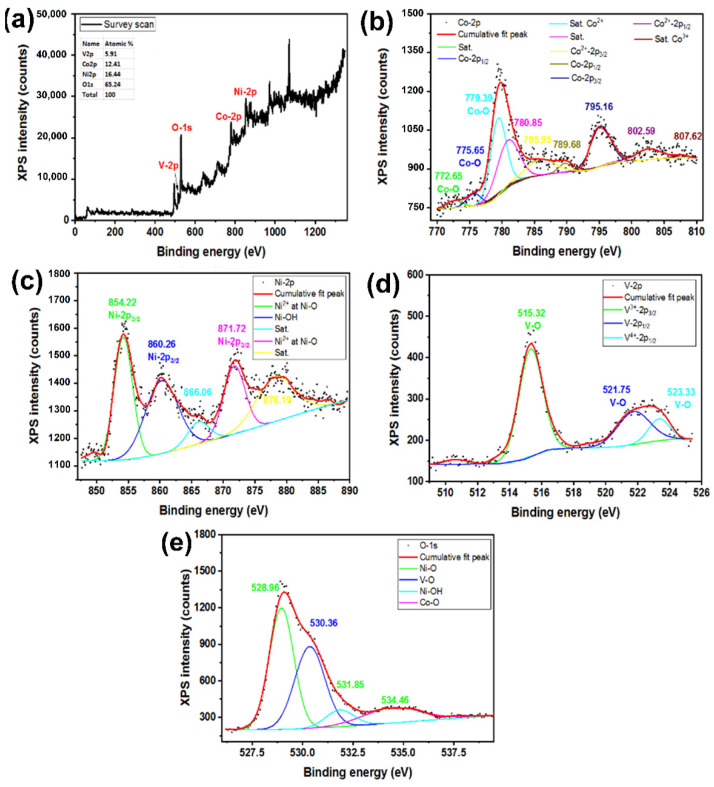
XPS of a Co*_x_*NiV*_y_*O*_z_* layer coated on a Ni foam substrate, (**a**) survey scan, (**b**) Co-2*p*, (**c**) Ni-2*p*, and (**d**) V-2*p*, and (**e**) O-1*s*.

**Figure 7 nanomaterials-12-01894-f007:**
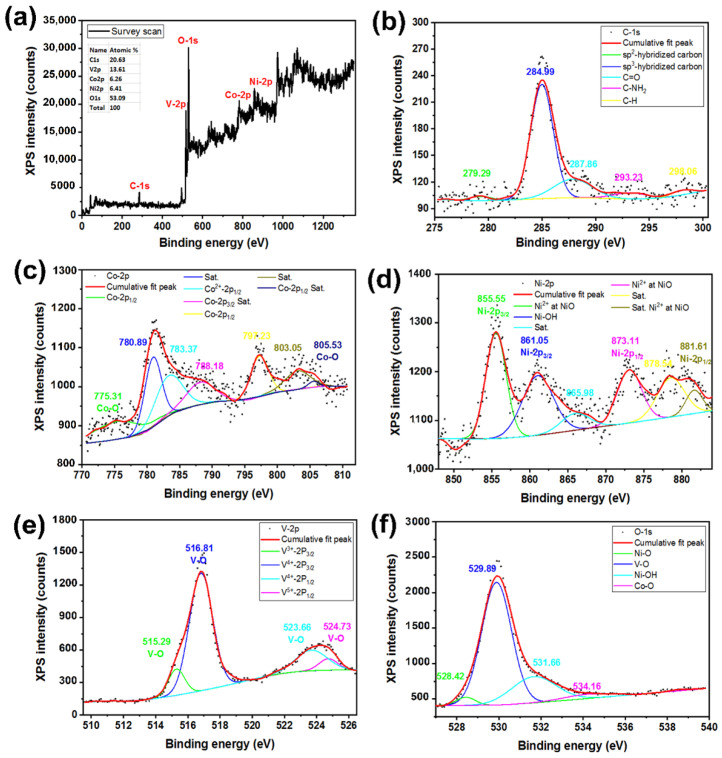
XPS of a Co*_x_*NiV*_y_*O*_z_* layer coated on a Ni foam@rGO substrate, (**a**) survey scan, (**b**) C-1*s*, (**c**) Co-2*p*, (**d**) Ni-2*p*, (**e**) V-2*p*, and (**f**) O-1*s*.

**Figure 8 nanomaterials-12-01894-f008:**
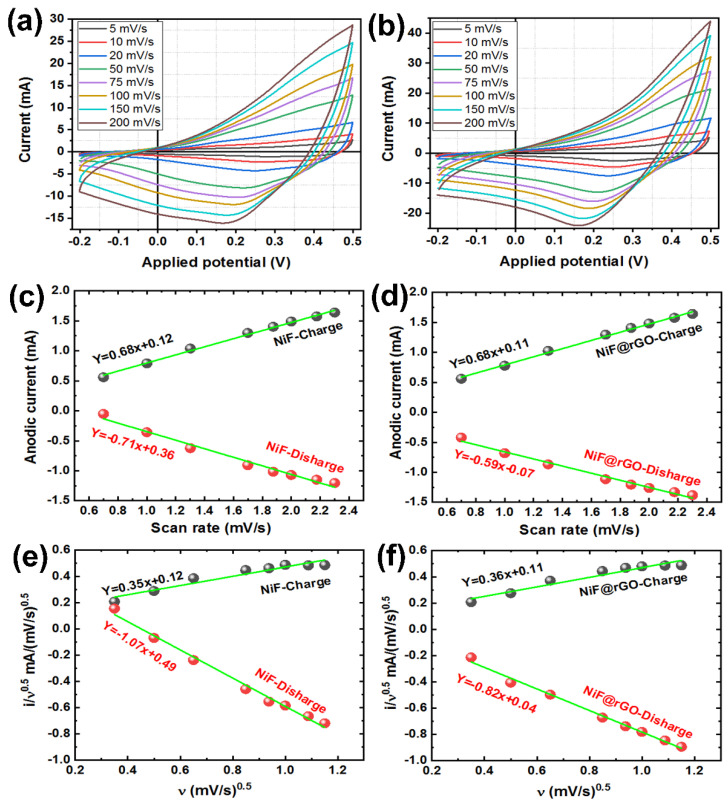
CV curves and the logarithmic dependence of current on the scan rate of a typical Co*_x_*NiV*_y_*O*_z_*-based electrode coated on (**a**,**c**,**e**) Ni foam and (**b**,**d**,**f**) Ni foam@rGO substrate tested in KOH electrolyte (1.0 M) at scan rates ranging from 5 to 200 mV/s.

**Figure 9 nanomaterials-12-01894-f009:**
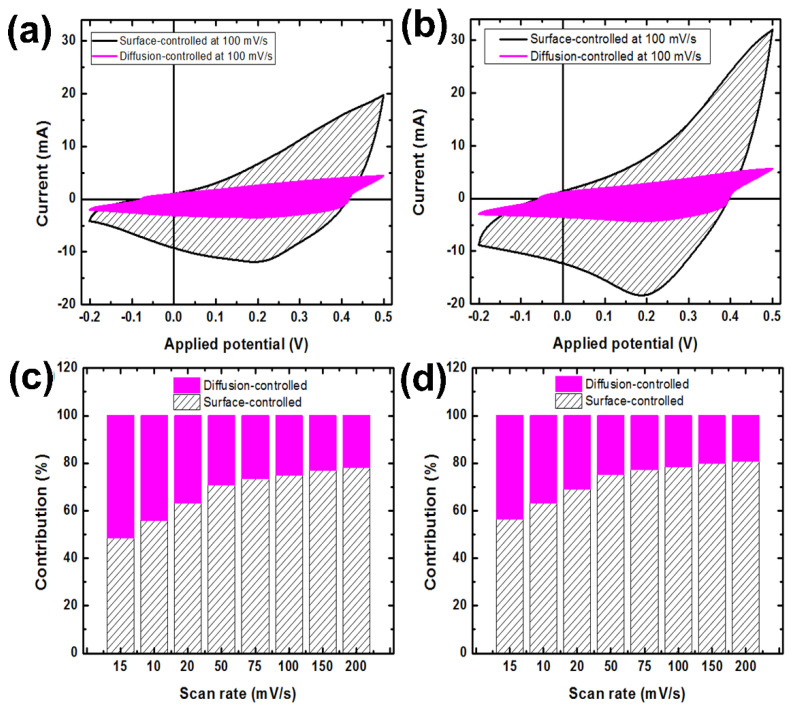
Typical CV curves and contribution of the different mechanisms on the capacitive behavior of the Co*_x_*NiV*_y_*O*_z_* electrode deposited on the (**a**,**c**) Ni foam and (**b**,**d**) Ni foam@rGO substrates.

**Figure 10 nanomaterials-12-01894-f010:**
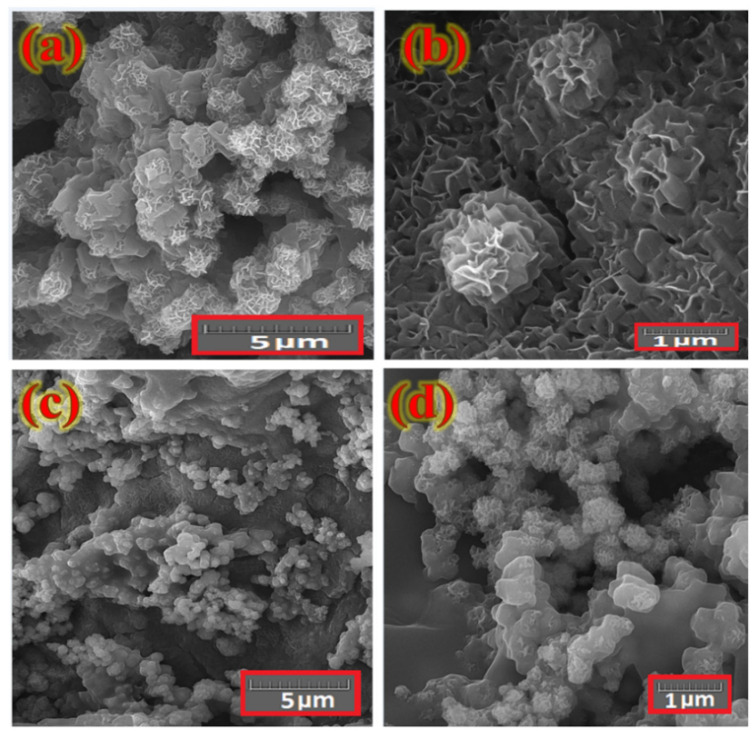
FESEM images of a typical interlaced Co*_x_*NiV*_y_*O*_z_* layer deposited on bare Ni foam (**a**,**b**), and Ni foam@rGO substrate (**c**,**d**) after 5500 consecutive charge–discharge cycles at 10 A g^−1^.

**Figure 11 nanomaterials-12-01894-f011:**
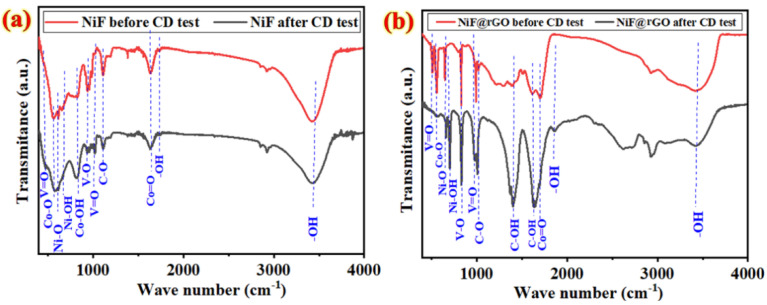
FTIR spectra of various samples coated on (**a**) Ni foam, (**b**) Ni foam@rGO substrate before and after electrochemical reduction.

**Figure 12 nanomaterials-12-01894-f012:**
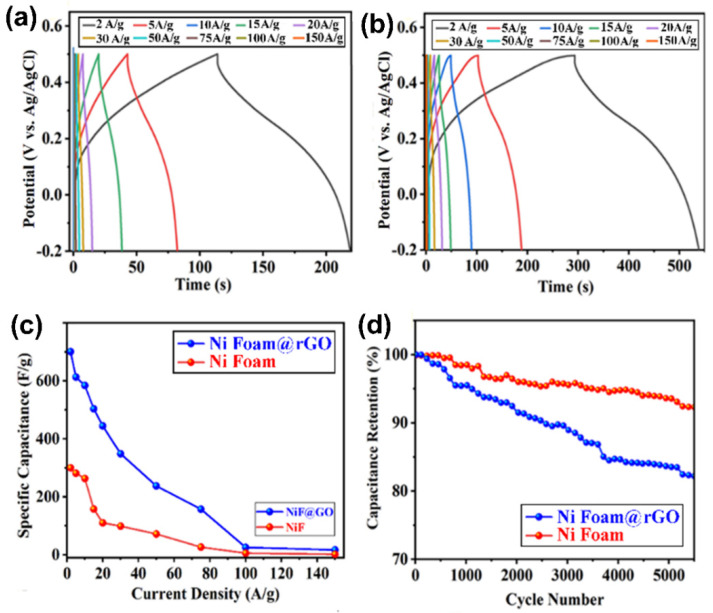
GCD curves of a typical Co*_x_*NiV*_y_*O*_z_* material deposited on various substrates. (**a**) Ni foam substrate, (**b**) Ni foam@rGO substrate at current density ranging from 2 to 150 A/g in KOH electrolyte (1.0 M). (**c**) Calculated specific capacitance, and (**d**) capacitance retention of Co*_x_*NiV*_y_*O*_z_* electrodes deposited on various substrates.

**Table 1 nanomaterials-12-01894-t001:** Coulombic efficiency of the Co*_x_*NiV*_y_*O*_z_* films deposited on the various substrates.

Sample Deposited On	Coulombic Efficiency (%)									
	2 A/g	5 A/g	10 A/g	15 A/g	20 A/g	30 A/g	50 A/g	75 A/g	100 A/g	150 A/g
Ni foam	92	93	90	99	100	105	102	100	75	30
Ni foam@rGO	83	84	88	94	98	101	99	101	108	60

**Table 2 nanomaterials-12-01894-t002:** Comparison between specific capacitance of the suggested electrode materials in the present research and similar reported references (all the reports used the Ni Foam as current collector).

Electrode Materials	Synthesis/Deposition Method	Specific Capacitance (F/g)	Current Density or Scan Rate (A/g)	Electrolyte	Ref.
MnO_2_	Layer by layer/electrodeposition	304.00	2 A/g	1 M Na_2_SO_4_	[76]
MnO_2_	electrodeposition	469.00	1 A/g	1 M Na_2_SO_4_	[77]
Co_3_O_4_/rGO	Hydrothermal	440.40	5 A/g	2 M KOH	[78]
Vö-V_2_O_5_ nano-Fiber	polymerization process of poly-aniline (PANI)	523.00	0.5 A/g	1 M Na_2_SO_4_	[79]
Mn-Ni-Co oxide	Hydrothermal	638.00	1 A/g	6 M KOH	[80]
V_2_O_5_	Electrodeposition	421.00	1 A/g	0.1 M H_2_SO_4_	[81]
Nickel Vanadium Oxide	Hydrothermal	297.00	3 A/g	1 M LiCl	[82]
Co_3_O_4_-NiO	Electrodeposition	687.00	0.5 A/g	2 M KOH	[83]
Cu(OH)_2_@rGO	Hydrothermal	602	2 A/g	6 M KOH	[84]
Co*_x_*NiV*_y_*O*_z_*	Electrodeposition	701.08	1 A/g	1 M KOH	This Work

## Data Availability

All data in the main text or Appendix A are available upon request.

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
