# Peer review of "Electrodeposition of CoxNiVyOz Ternary Nanopetals on Bare and rGO-Coated Nickel Foam for High-Performance Supercapacitor Application"

_nanomaterials, 2022, doi:10.3390/nano12111894_

Round 1

Reviewer 1 Report

The conventional energy storage system such as Li-ion rechargeable batteries fail to translate the necessary power to the system due to the limited ionic diffusion kinetics. On the other hand, the supercapacitors are unable to store the necessary energy owing to the non-Faradaic charge storage mechanism, though it offers a high-power capability (>10 kW kg–1). Therefore, the authors have presented in the current work, mixed metal oxides, which leads to the exploration of different kinds of charge storage systems that can offer both high-energy and high-power capability through ternary composites. Ni foam has been utilized as a scaffold for the uniform growth of electrode materials and is also used as the support of a current collector. A facile, cost-effective one-step hydrothermal process has been employed to synthesize the ternary electrode.

The characterization is neat and both physical and electrochemical analyses are well balanced. Nevertheless, the work in its present form is not publishable and needs revisions before rendering a final decision.  The role of nano-petals and their charge transfer process, and the features of redox behavior in aqueous electrolytes need to be explained. Capacitors are a hot topic researched area, and the prior work needs to be promptly referenced to benchmark the obtained capacitance values.

The following points need to be considered.

Major

  • The distinctive combination of Co; Ni; and V has been adopted. On what basis these cations were adopted and explain the synergistic contribution from individual components
  • The intercalation, conversion, and redox reaction mechanisms during the electrochemical charge-discharge process should be demonstrated.
  • Section 2.4.3 The reported electrodeposition of nanocomposites is widely reported by Minakshi and Biswal et al for energy storage devices Please refer to their work and bring the merits of electrodeposition.
  • Page 7; lines 228 – 230 (not stick tightly together) – is it good or bad news?
  • Section 3.2 The dependence of the current density on the scan rate in Figure 8 needs to be demonstrated.
  • Section 3.3 Please provide the ratios of surface-controlled and diffusion-controlled capacitance of the electrode behavior shown in Fig. 11a-b.
  • Section 3.3 A similar oxide material reported for supercapacitors must be benchmarked in the manuscript (such as Ceramics International doi.org/10.1016/j.ceramint.2022.03.266; RSC Advances 9 (2019) 26981).
  • What is the coulombic efficiency of the charge-discharge cycle?
  • Do the V cation dissolve due to the reaction with OH- which may prevent the electrolyte ions from entering the ternary composite, and the dissolution of the V reduces the actual effective energy storage space. Is this the fact behind the deteriorating values shown in Fig. 11 c-d?

Minor

  • Page 13, line 353 “principle of electrostatic charge storage” is the authors sure about this? Maybe electrochemical double layer rather electrostatic, Double-check.
  • 11c, inset is not readable.
  • Figure 11 a-b y-axis unit should read as “potential (V) versus (Ag/AgCl)”. Is that correct?
  • The term “NiF” can be misrepresented as Ni Fluoride rather than Ni Foam. Be mindful, that readers may get the wrong interpretation.
  • Page 2; line 70 what is the impact of structural defects? Either cite the previous work or provide details. Otherwise, it is dubious.

Reviewer 2 Report

The current manuscript demonstrates the supercapacitor application of Co-Ni-V ternary oxide grown on Ni foam and rGO-coated Ni foam. The manuscript delivers significant data for considering the publication in this journal. Therefore, I recommend acceptance of this manuscript after minor revision. My comments are given below -

  1. The introduction part should be enlarged.
  2. Why rGO-coated Ni foam gives lower cyclic stability than bare Ni foam-coated electrode material? The authors should explain this in their manuscript. 
  3. The authors should increase the resolution of the figures like Fig. 11.
  4. In Figure 8, the scan rates should be expressed in terms of the unit mV/s. Similarly, in Figure 11, the current densities should be expressed in terms of A/g units.
  5. The probable electrochemical reactions should be provided in the manuscript.
  6. The obtained electrochemical data should be compared with other mixed metal oxides grown on Ni foam (previously published). A separate comparison table should be provided.   

Author Response

The current manuscript demonstrates the supercapacitor application of Co-Ni-V ternary oxide grown on Ni foam and rGO-coated Ni foam. The manuscript delivers significant data for considering the publication in this journal. Therefore, I recommend acceptance of this manuscript after minor revision. My comments are given below:

  1. The introduction part should be enlarged.

We would like to thank the reviewer for this constructive comment. In order to clarify the importance of supercapacitor, nickel foam substrate, and give insight to readers about graphene covering effect, new sentences were added according to the suggestion of referee.

New related references 13-18 and 24-27 were also added. We believe that the introduction is now acceptable for readers.

  1. Why rGO-coated Ni foam gives lower cyclic stability than bare Ni foam-coated electrode material? The authors should explain this in their manuscript. 

The bellow explanations were added in revised manuscript (page 17 lines: 467-488) to answer reviewer’s question:

Despite higher specific capacitance, the CoxNiVyOz based electrode material elec-trodeposited on two dimensional rGO coated Ni Foam failed to achieve the expected stability, mainly due to the high contact resistance between the graphene and Ni Foam substrate. To deal with the similar issue, researchers have made attempts to use other deposition methods such as chemical vapor deposition (CVD) [66], vacuum filtration deposition (VFD) [67], hydrothermal [68], and electrophoretic deposition (EPD) method [69, 70] for generating binder-free electrodes where the graphene material stick tightly on the metal substrate. Another proposed way to solve this problem is assembling the graphene sheets into a three-dimensional (3D) framework including aerogels, foams and sponges in graphene-based electrodes [71- 74]. The novelty of this study is to introduce the electrodeposited CoxNiVyOz ternary nanopetals as a new electrode material which provides high performance supercapacitor by enough channels for ion storage and migration. The obtained specific capacitance of the present research was compared with similar reports presented in Table 2. By comparing our results with others using metal oxide-based electrode materials fabricated with different deposition methods (some examples are listed in Table 2), our proposed electrode material showed good potential to use in supercapacitor. Electrochemical deposition as a low-cost, simple controlled, and low temperature growth technique is usually carried out in normal laboratory conditions without requiring a vacuum system [24-26, 74]. In order to use an easy and rapid growth process, we didn’t emphasize on other deposition method or preparation of three dimensional graphene based substrate to prevent the graphene sheet from restacking and keep their specific surface area to increase the stability of prepared electrode.

  1. The authors should increase the resolution of the figures like Fig. 11.

It is done in the revised manuscript.

  1. In Figure 8, the scan rates should be expressed in terms of the unit mV/s. Similarly, in figure 11, the current densities should be expressed in terms of A/g units.

It is done in the revised manuscript.

  1. The probable electrochemical reactions should be provided in the manuscript.

Redox reaction mechanisms and related descriptions were added according to the suggestion of referee in page 11 lines 303-315.

  1. The obtained electrochemical data should be compared with other mixed metal oxides grown on Ni foam (previously published). A separate comparison table should be provided.

We would like to thank the reviewer for this constructive comment. Accordingly, the results of early studies on metal oxide electrode materials coated on Ni foam are summarized in Table 2 for comparison.

Round 2

Reviewer 1 Report

I went through the revised version of the manuscript. The authors have fairly addressed my queries satisfactorily.